# Prevalence of Glaucoma and Its Bayesian Risk Estimation Model Using Common Determinants in the Adult Population of Hail Region, Saudi Arabia

**DOI:** 10.3390/healthcare12232423

**Published:** 2024-12-03

**Authors:** Abrar Ali, Zaki Aqeel Alshammari, Fahad Mohammed Alshomer, Njoud Saleh Alanezi, Othman Mohammad Alassaf, Sarah Khalid Albarrak, Sami Ibrahim Alzabni, Khaled Homod Almozaini, Solaiman Ismail Alamer, Nabeel Shalaby, Mohd Saleem, Azharuddin Sajid Syed Khaja

**Affiliations:** 1Department of Ophthalmology, College of Medicine, University of Hail, Hai 55476, Saudi Arabia; z.allogan@uoh.edu.sa (Z.A.A.); n.shalabi@uoh.edu.sa (N.S.); 2Department of Optometry, Hail Health Cluster, College of Applied Medical Science, King Saud University, Riyadh11433, Saudi Arabia; fa-m-s@hotmail.com; 3College of Medicine, University of Hail, Hail 55476, Saudi Arabia; s201616395@uoh.edu.sa (N.S.A.); othman7552@gmail.com (O.M.A.); saruna7070@hotmail.com (S.K.A.); sami_alzabni@hotmail.com (S.I.A.); s201803116@uoh.edu.sa (K.H.A.); slymanalaamer@gmail.com (S.I.A.); 4Department of Pathology, College of Medicine, University of Hail, Hail 55476, Saudi Arabia; m.saleem@uoh.edu.sa

**Keywords:** glaucoma, glaucoma prevalence, Hail, Bayesian Risk Estimation, risk factors, age-related glaucoma, diabetes and glaucoma, hypertension and eye health

## Abstract

Background/Objectives: Glaucoma is a global health concern, with an anticipated rise from 64.5 million cases in 2014 to 112 million by 2040. In Saudi Arabia, it contributes to 5.7% of visual impairment cases. Early detection through routine eye exams is crucial, as glaucoma often progresses asymptomatically, leading to irreversible vision loss if left untreated. The present study aims to determine the prevalence of glaucoma in the Hail region of Saudi Arabia. Methods: For this cross-sectional study, a sample of 200 participants underwent demographic assessment, and a Bayesian Risk Estimation Model was employed to analyze determinants such as age, gender, education, and comorbidities. Results: The cross-sectional study in the Hail region of Saudi Arabia involving 9407 outpatients revealed a glaucoma prevalence of 2.1%, with key factors influencing glaucoma risk identified, including age (60–69 years with a 43.38% chance), illiteracy (22.58% chance), and comorbidities such as diabetes mellitus (16.10% chance) and cataract (15.40% chance). Conclusions: In conclusion, the study in the Hail region highlights a 2.1% prevalence of glaucoma, emphasizing the significant impact of age, education, and comorbidities on glaucoma risk. These findings underscore the importance of targeted interventions for at-risk populations to enhance glaucoma management and prevention efforts.

## 1. Introduction

Glaucoma is a condition of increased intraocular pressure in the eye that may progress to a loss of vision. This results in a characteristic optic nerve head appearance on fundoscopic examination and a corresponding progressive vision loss. Four general categories of adult glaucoma exist: primary open-angle (POAG) and angle-closure, and secondary open-angle and angle-closure glaucoma [1].

In 2014, there were 64.5 million cases of glaucoma globally, and it is anticipated that this figure will rise to 112 million by the year 2040. The worldwide prevalence of glaucoma was estimated to be 3.54% [2]. In Saudi Arabia, glaucoma accounted for 5.7% of all causes of visual impairment, ranking as one of the top three major contributors to visual impairment [3]. The primary risk factor for glaucoma is elevated intraocular pressure (IOP), although normal-tension glaucoma can occur. Glaucoma is classified based on various factors, including onset (congenital or acquired), anatomy [primary open-angle glaucoma (POAG) or angle-closure glaucoma (ACG)], and etiology (primary or secondary) [4]. POAG, the most common type, is influenced by factors such as age, elevated IOP, ethnicity, family history, and hyperopia. Certain populations are at higher risk for specific subtypes; for instance, individuals of Asian descent and older patients are at increased risk for ACG, while congenital glaucoma, particularly primary congenital glaucoma (PCG), has an especially high incidence in Saudi Arabia (1 in 2500) [4]. Known risk factors for glaucoma include age, gender, and genetic predisposition.

This activity illustrates the evaluation and management of glaucoma and explains the role of the interprofessional team in improving care for patients with this condition.

Primary open-angle glaucoma typically progresses asymptomatically until advanced stages, making routine eye exams crucial for early detection. On the contrary, acute angle-closure glaucoma manifests abruptly, leading to a swift decline in vision, accompanied by symptoms like corneal edema, eye pain, headache, nausea, and vomiting. Secondary glaucoma is associated with prior eye injuries or diseases, causing elevated IOP and optic neuropathy [5]. Normal-tension glaucoma mirrors POAG but with normal IOP readings, potentially linked to a vascular autoregulatory defect. While congenital and juvenile variants exist, the four main glaucoma types primarily affect individuals over 40, with a potential genetic predisposition [6].

The pathophysiology involves the ineffective drainage of aqueous humor, leading to increased resistance in the trabecular meshwork or decreased blood flow to optic nerve fibers. As the disease progresses, peripheral vision is gradually lost, often unnoticed by affected individuals until central vision is compromised. Secondary open-angle glaucoma can result from various causes, including laser surgery, neovascularization, or the use of certain medications [7].

The diagnosis involves comprehensive eye examinations, visual field testing, tonometry to measure IOP, and additional imaging techniques. While no gold standard test for glaucoma exists, the American Academy of Ophthalmology recommends routine eye exams for individuals with risk factors [8]. Glaucoma management aims to control IOP to prevent further vision loss. Medications, laser procedures, and surgeries are employed based on the type and severity of the condition. Patient education emphasizes the importance of regular eye exams, as early detection and intervention are crucial for preserving vision [9]. Regular monitoring and adherence to treatment plans contribute to a more favorable prognosis, preventing the irreversible vision loss associated with untreated glaucoma. This study aimed to investigate the prevalence of glaucoma in the Hail region of Saudi Arabia, emphasizing the impact of late diagnosis on vision loss and providing insights for improved healthcare strategies. The rationale behind this study was rooted in the critical need to address late glaucoma diagnosis, a significant contributor to vision loss. By estimating the prevalence of glaucoma in Hail, Saudi Arabia, and establishing its relationship with vision impairment, our study aimed to provide valuable insights for future interventional research. The overarching goal was to position this study as a reference point for regional healthcare strategies, contributing to the overall improvement of glaucoma management and prevention efforts.

This study uniquely integrates a Bayesian Risk Estimation Model to evaluate glaucoma determinants, providing a probabilistic perspective on risk factors, and focuses specifically on the Hail region of Saudi Arabia, addressing a regional gap in the literature with comprehensive epidemiological and clinical insights.

## 2. Materials and Methods

### 2.1. Study Design and Period

It is a retrospective study conducted in two major centers of the Hail region in Saudi Arabia between September 2022 and January 2023.

### 2.2. Study Ethical Approval

This study was approved by the scientific research ethical committee of the University of Hail (H-2022-339) and by the institutional review board registration of the Hail region (KACS, KSA, H-08-L-074), IRB Log No. (2022-72).

### 2.3. Study Population and Sample Size

The sample size was calculated using Daniel’s formula, based on the expected prevalence of glaucoma in the Saudi Arabian region, which was estimated at 5.7% [3]. With a 95% confidence level, a study power of 90%, and an anticipated 7.5% loss to follow-up, the required sample size was determined to be 9323. Therefore, the initial sample size was set at 9407, in alignment with the available resources and in close proximity to the calculated figure.

So, the initial sample size consisted of 9407 outpatients from the Hail region, specifically chosen to assess the prevalence and risk factors associated with glaucoma. The sample collection was multifaceted, involving a purpose-designed, pre-validated, and translated self-administered form to gather demographic and relevant health information. Two hundred (200) glaucoma patients were targeted for inclusion in the study for further analysis.

### 2.4. Data Collection and Statistical Analysis

The sample collection process involved the review of files and charts of both old and new patients attending two medical centers in the Hail region. This study aimed to identify cases of glaucoma and assess the specific criteria for diagnosis, including glaucoma types. The Statistical Package for Social Sciences (SPSS) version 23 was employed for statistical analysis, facilitating the synthesis of descriptive statistics to characterize the demographic profile of the study population.

Moreover, a Bayesian Risk Estimation Model was applied to estimate the effect of determinants on glaucoma. This model incorporated factors such as age, gender, educational status, and comorbidities like diabetes mellitus (DM), hypertension, asthma, cardiovascular disease (CVD), and cataract. The results were analyzed using descriptive statistics and making comparisons among various groups. Categorical data were summarized as proportions and percentages (%) while discrete data were summarized as the mean ± SD.

Bayes’ Formula: Bayes’ theorem describes the probability of occurrence of an event related to any condition. It is also considered for the case of conditional probability. Bayes’ theorem is a mathematical formula for calculating conditional probability in probability and statistics. In other words, it is used to figure out how likely an event is based on its proximity to another.
$$P\left(\frac{WD}{RF}\right)=\frac{P\left(\frac{RF}{WD}\right)\cdot P\left(WD\right)}{P\left(RF\right)}$$

## 3. Results

### 3.1. Epidemiological Profile of Glaucoma

In order to attain the goal of enrolling 200 glaucoma patients, the present study was conducted on 9407 outpatients (OPD), revealing a glaucoma prevalence of 2.1% (95% CI 1.7–2.5%, Table 1).

### 3.2. Distribution of Cases According to Demographic Profile

The present study population’s demographic characteristics were comprehensively examined, providing valuable insights into the distribution of key variables (Table 2). In terms of age, participants were well represented across various age groups, with 25.5% falling in the 40–49 years range, 34.5% in the 50–59 years range, 27.5% in the 60–69 years category, 8.5% in the 70–79 years group, and 4.0% aged 80 years and above. The mean age was calculated as 57.63 ± 9.92 years (95% confidence interval (CI) = 56.3% to 59.0%). Gender distribution indicated a relatively balanced representation, with 52.5% male and 47.5% female, within 95% CIs of 45.6–59.4% and 40.6–54.4%, respectively. Educational status revealed that most participants were literate (97.5%, 95% CI = 95.3–99.7%), with only a small percentage (2.5%, 95% CI = 0.3–4.7%) being illiterate.

### 3.3. Distribution of Cases According to Glaucoma and Clinical Features

The distribution of cases based on glaucoma and its clinical features provides a comprehensive overview of the prevalence and characteristics within the study population (Table 3). Regarding patient complaints, various symptoms were reported, with 10.0% experiencing blurry vision, 25.5% reporting decreased vision, and 7.5% noting headaches or pain. Additionally, 27.0% developed glaucoma during follow-up, 5.0% were diagnosed during a routine checkup, 1.5% reported loss of vision, 1.0% mentioned black spots, and 2.5% already had a known history of glaucoma. The category labeled as ‘Other/NA’ accounted for 20.0% of cases (95% CI = 14.5–25.5%). Concerning glaucoma types, the overwhelming majority (99.0%) had primary glaucoma, while only 1.0% were diagnosed with secondary glaucoma (95% CI = 0.0–2.4%). Regarding the duration of glaucoma, 72.0% of cases were newly diagnosed (95% CI = 65.8–78.2%). The remaining cases were distributed across different time intervals, including 2.0% with a duration of less than or equal to 1 year, 6.0% between 2 and 5 years, 14.5% between 6 and 9 years, and 5.5% with a duration exceeding ten years. These findings offer valuable insights into the clinical presentation and characteristics of glaucoma cases.

### 3.4. Distribution of Cases According to Comorbidities Among Glaucoma Patients

Table 4 provides a comprehensive breakdown of the comorbidities among glaucoma patients, highlighting the prevalence of diabetes mellitus, hypertension, cataract, and other conditions. This information underscores the need for integrated care that addresses both ocular and systemic health factors in glaucoma patients. Among the identified comorbidities, DM was the most common, affecting 46.5% of glaucoma patients (95% CI = 39.6–53.4%). Hypertension was the second most prevalent comorbidity, observed in 20.5% of cases (95% CI = 14.9–26.1%). Asthma and cardiovascular disease (CVD) were less frequently reported, affecting 2.0% (95% CIs = 0.1–3.9%) and 1.0% (95% CIs = 0.0–2.4%) of glaucoma patients, respectively. Cataract, another common ocular condition, was identified in 11.0% of glaucoma patients (95% CI = 6.7% to 15.3%). These findings underscore the importance of considering comorbidities in the overall management and care of individuals diagnosed with glaucoma.

### 3.5. Estimation of Effect of Determinants over Glaucoma Using Bayes’ Formula

Estimating the effect of determinants on glaucoma using Bayes’ formula provides a probabilistic perspective on the likelihood of developing glaucoma based on various factors (Table 5, Figure 1). The analysis considers the proportion of each determinant in the general population and calculates the percentage chances of glaucoma for different subgroups. Age is a significant determinant, with higher percentages of glaucoma associated with increasing age. For individuals aged 60–69 years, the chances of glaucoma are notably higher at 43.38%, compared to 8.42% for those aged 40–49 years. Similarly, the gender variable reveals that males have a 5.09% chance of glaucoma, whereas females have a slightly higher probability of 6.30%. Education plays a role in glaucoma likelihood, with illiterate individuals having a 22.58% chance compared to 5.49% for the literate population. Comorbidities such as DM and hypertension significantly influence the chances of glaucoma, with probabilities of 16.10% and 4.59%, respectively. Asthma, CVD, and cataract also contribute to the probability of glaucoma, with chances ranging from 1.02% to 15.40%. These estimations highlight the importance of considering multiple factors when assessing the risk of glaucoma, providing a more nuanced understanding of the disease’s determinants in the studied population.

### 3.6. Descriptive Summary of Seven Most Common Factors of Glaucoma

The descriptive summary outlines the percentage chances of developing glaucoma associated with the seven most common factors in the studied population (Table 6). The factors include age categories, educational status, and specific comorbidities.

According to the findings, individuals in the age group of 60–69 years have the highest probability of developing glaucoma, with a substantial percentage chance of 43.38%. The likelihood of glaucoma remains significant in the age category of 70–79 years, with a notable percentage chance of 35.52%. Even among the elderly population, those aged 80 years and above exhibit a substantial chance of glaucoma, with a percentage of 37.33%. Additionally, the age group of 50–59 years shows a considerable chance of glaucoma, with a percentage of 22.78%. Educational status appears to influence the likelihood of glaucoma, as illiterate individuals have a significant percentage chance of 22.58%. Individuals with diabetes mellitus exhibit a notable chance of developing glaucoma, with a percentage of 16.10%. The presence of cataract contributes to the chances of glaucoma, with a substantial percentage of 15.40%.

This descriptive summary emphasizes the impact of age, education, and specific comorbidities on the probability of glaucoma in the studied population. It provides valuable insights for healthcare professionals and policymakers to target interventions and preventive measures for individuals with these identified risk factors.

## 4. Discussion

In this cross-sectional study conducted in the Hail region of Saudi Arabia, the primary aim was to estimate the prevalence of glaucoma, addressing a significant knowledge gap due to the absence of prior regional research. The study analyzed data from 9407 outpatients and found a glaucoma prevalence of 2.1% (95% CI: 1.7–2.5%), with a mean participant age of 57.63 years. The highest prevalence was observed in individuals aged 50–59 years (34.5%), and the gender distribution was nearly balanced at 52.5% male and 47.5% female.

Age was a notable risk factor, with a 43.38% chance of glaucoma in participants aged 60–69. Gender differences were modest, with slightly higher risk in females (6.3%) compared to males (5.09%). Comparisons with other studies highlight variability in glaucoma prevalence across regions and populations. Studies from other parts of Saudi Arabia [10,11,12] and globally [13,14,15] show differences due to demographic, ethnic, and environmental factors, with age and gender frequently impacting glaucoma risk. For instance, studies in the Jazan [10] and Jeddah [11] regions indicated age and gender variations, while international studies demonstrated higher prevalence among specific ethnic groups, such as African-Caribbean populations and East Asians [13,14]. This variation underscores the importance of contextualized regional studies in understanding glaucoma epidemiology and informing targeted health interventions.

In comparing our findings with those of other studies, such as [16], which observed a significant increase in glaucoma prevalence with age, we found a similar trend, with higher prevalence among older age groups. However, while [16] reported a seven-fold increase up to age 80–84 in females and age 85–89 in males, our study did not capture as steep an increase in these age brackets. This difference could be attributed to regional and demographic variations between populations, as well as our study’s limited sample size in the oldest age groups. Additionally, our study observed a relatively balanced prevalence between males and females, whereas other studies noted a significantly higher prevalence in females at specific age ranges. These variations might reflect differences in genetic, environmental, or healthcare access factors across regions. Future studies with larger and more diverse samples across age ranges could clarify these demographic influences on glaucoma prevalence in Hail and similar regions.

In this study, the majority of participants (97.5%) were literate, with illiteracy associated with a higher glaucoma risk (22.58% vs. 5.49% in literate individuals). This finding underscores the influence of socioeconomic factors on glaucoma risk. Previous studies, like Almarzouki et al. (2021), found a positive correlation between higher educational levels and awareness of glaucoma, highlighting education’s role in understanding and preventing this condition [17].

The study also revealed that 10.0% of participants experienced blurry vision, 25.5% had decreased vision, and 7.5% reported headaches or eye pain. Routine checkups were critical, identifying 5.0% of glaucoma cases. Comparatively, Al-huthaili et al. (2023) noted similar symptoms in their participants, while Hu et al. (2014) found blurry vision and the need for more light were common complaints among those with primary open-angle glaucoma [18,19].

Most cases in this study (99.0%) were primary glaucoma, with only 1.0% being secondary. Approximately 72.0% were newly diagnosed cases, emphasizing early detection’s importance. Comparisons with international and regional studies, including those by Douglass et al. (2023) and Helayel et al. (2021), show a similar trend of primary glaucoma predominance across different populations [20,21].

Diabetes mellitus (DM) emerged as the most prevalent comorbidity (46.5%) in glaucoma patients, followed by hypertension (20.5%), reinforcing the association between systemic health conditions and glaucoma. Previous studies, such as those by Alamri et al. (2022) and Alzuhairy et al. (2018), also highlighted the high prevalence of DM and hypertension among glaucoma patients [22,23]. Similar findings in South Indian and older adult populations further underscore the interconnectedness of systemic health and ocular conditions, with diabetic retinopathy also posing a significant risk to eye health [24,25]. These findings stress the need for holistic management of glaucoma patients, accounting for both ocular and systemic health factors.

In this study, cataracts were found in 11.0% of glaucoma patients, highlighting the prevalence of co-existing eye conditions. This finding reflects the complexity of managing multiple ocular diseases concurrently. Similarly, a study by Ahmedhussain et al. (2023) in the western region of Saudi Arabia reported an 8.3% cataract prevalence, emphasizing the condition as a notable concern in Saudi Arabia [26].

These findings underscore the importance of holistic healthcare strategies that address co-existing eye conditions and consider regional demographic factors. By focusing on such integrated approaches, this study contributes valuable insights for improving comprehensive management and interventions to enhance overall eye health outcomes among individuals with glaucoma.

### Limitations of the Present Study

The study’s limitations include the retrospective and cross-sectional design, which offers a strong foundation but could be expanded with prospective, longitudinal studies to capture disease progression over time. The study does not employ advanced diagnostic tools like optical coherence tomography. Moreover, secondary glaucoma cases are underrepresented (1%), and potential confounding factors, such as lifestyle and genetic predisposition, are not fully explored. Additionally, a larger and more diverse sample, including underrepresented populations and broader risk factors, could further enrich the findings and ensure even greater applicability.

## 5. Strengths of the Present Study and Conclusions

The present study provides valuable insights into the prevalence and risk factors associated with glaucoma in the Hail region of Saudi Arabia, a previously underexplored area in glaucoma research. It utilizes a Bayesian Risk Estimation Model to assess glaucoma risk based on multiple factors, providing insights beyond traditional methods. With a glaucoma prevalence of 2.1% among 9407 outpatients, the findings underscore the significant impact of age, illiteracy, and comorbidities such as diabetes and hypertension on glaucoma risk. Individuals aged 60–69 years and those with lower educational levels are particularly vulnerable, highlighting the need for targeted screening and preventive interventions. The study emphasizes the importance of early detection and comprehensive management strategies, particularly for high-risk populations, to prevent vision loss and improve patient outcomes in glaucoma care. By comparing regional and global data, the study delivers a localized perspective with broad implications for glaucoma management.

## Figures and Tables

**Figure 1 healthcare-12-02423-f001:**
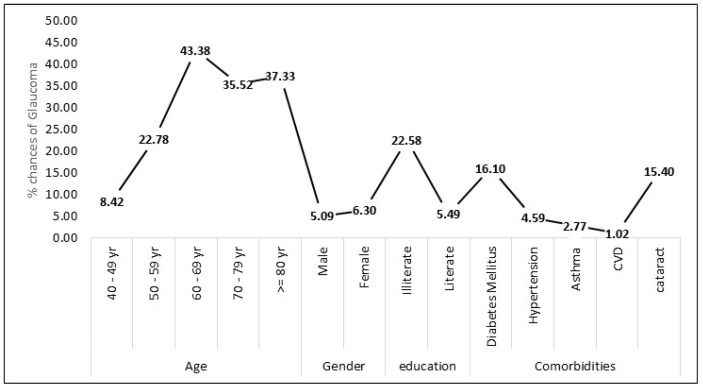
Estimation of effect of determinants over glaucoma using Bayes’ formula.

**Table 1 healthcare-12-02423-t001:** Distribution of cases according to prevalence of glaucoma.

Disease Status	No.	%	95% CI
Glaucoma	200	2.1%	1.7–2.5%
No Glaucoma	9207	97.9%	
Total	9407	100.0%	100.0%

**Table 2 healthcare-12-02423-t002:** Distribution of cases according to demographic profile.

Variables	No.	%	95% CI
**Age (Mean ± SD)**	57.63 ± 9.92 yrs.		56.3–59.0%
40–49 years	51	25.50%	19.5–31.5%
50–59 years	69	34.50%	27.9–41.1%
60–69 years	55	27.50%	21.3–33.7%
70–79 years	17	8.50%	4.6–12.4%
≥80 years	8	4.00%	1.3–6.7%
**Gender**			
Male	105	52.50%	45.6–59.4%
Female	95	47.50%	40.6–54.4%
**Education**			
Illiterate	5	2.50%	0.3–4.7%
Literate	195	97.50%	95.3–99.7%

**Table 3 healthcare-12-02423-t003:** Distribution of cases according to glaucoma diagnosis and clinical presentation.

Clinical Presentation/Diagnostic Context	No.	%	95% CI
**Patient-Reported Symptoms**			
Blurry vision	20	10.00	5.8–14.2%
Decreased vision	51	25.50	19.5–31.5%
Headache/Pain	15	7.50	3.8–11.2%
**Diagnostic context**			
Developed during follow-up	54	27.00	20.8–33.2%
Diagnosed in routine checkup	10	5.00	2.0–8.0%
Other/NA	40	20.00	14.5–25.5%
**Glaucoma type**			
Primary	198	99%	97.6–100%
Secondary	2	1%	0.0–2.4%
**Duration of the disease**			
Newly diagnosed	144	72.00%	65.8–78.2%
≤1 year	4	2.00%	0.1–3.9%
2–5 year	12	6.00%	2.7–9.3%
6–9 year	29	14.50%	9.6–19.4%
≥10 year	11	5.50%	2.3–8.7%

**Table 4 healthcare-12-02423-t004:** Distribution of cases according to comorbidities among glaucoma patients.

Comorbidities	No.	%	95% CI
Diabetes Mellitus	93	46.5%	39.6–53.4%
Hypertension	41	20.5%	14.9–26.1%
Asthma	4	2.0%	0.1–3.9%
Cardiovascular Disease	2	1.0%	0.0–2.4%
Cataract	22	11.0%	6.7–15.3%

**Table 5 healthcare-12-02423-t005:** Distribution of cases according to glaucoma and clinical features.

Variable	No.	%	Proportion in General Population	Chances of Glaucoma (%)
**Age**				
40–49 years	51	25.5	16.95	8.42
50–59 years	69	34.5	8.48	22.78
60–69 years	55	27.5	3.55	43.38
70–79 years	17	8.5	1.34	35.52
≥80 years	8	4	0.6	37.33
**Gender**				
Male	105	52.5	57.78	5.09
Female	95	47.5	42.22	6.3
**Education**				
Illiterate	5	2.5	0.62	22.58
Literate	195	97.5	99.38	5.49
**Comorbidities**				
Diabetes Mellitus	93	46.5	16.17	16.1
Hypertension	41	20.5	25	4.59
Asthma	4	2	4.05	2.77
CVD	2	1	5.5	1.02
Cataract	22	11	4	15.4

Source: Unstats.un.org.

**Table 6 healthcare-12-02423-t006:** Descriptive summary of seven most common factors of glaucoma.

Variable	% Chances of Glaucoma
Age 60–69 years	43.38
Age 70–79 years	35.52
Age ≥ 80 years	37.33
Age 50–59 years	22.78
Illiterate	22.58
Diabetes Mellitus	16.10
Cataract	15.40

## Data Availability

The data generated and analyzed during the current study are provided within the manuscript.

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
