# Peer review of "Prevalence of Glaucoma and Its Bayesian Risk Estimation Model Using Common Determinants in the Adult Population of Hail Region, Saudi Arabia"

_healthcare, 2024, doi:10.3390/healthcare12232423_

Round 1
Reviewer 1 Report
Comments and Suggestions for Authors
This study reported a 2.1% prevalence of glaucoma in the Hali region of Saudi Arabia.
Why did the authors select a sample size of 200 glaucoma patients?
The specific types of glaucoma should be clarified. It is unclear whether the identified risk factors were associated with primary angle-closure glaucoma, potentially induced by factors such as cataract or age, or if they were linked to primary open-angle glaucoma through other mechanisms.
The Discussion section contains redundant information and could be condensed by half for better clarity.
Author Response
Dear Editor,
Healthcare
MDPI
We appreciate the reviewers' thoughtful and constructive comments, which have significantly improved the quality and clarity of our manuscript " Prevalence of Glaucoma and its Bayesian Risk Estimation Model Using Common Determinants in the Adult Population of Hail Region, Saudi Arabia." We have carefully considered and addressed each comment comprehensively, incorporating additional analyses and clarifications where necessary. Below, we provide detailed responses to all comments, outlining the revisions made to the manuscript to meet the reviewers' expectations. We hope that our responses and revisions align with the reviewers' suggestions and enhance the overall impact of the study.
Best,
Dr. Azharuddin Sajid Syed Khaja
Response to Reviewers' comments
Reviewer-1
- Why did the authors select a sample size of 200 glaucoma patients?
Response: We selected a sample size of 200 glaucoma patients to achieve a robust yet manageable representation of the glaucoma population in the Hail region. This sample size allows for adequate statistical power to identify prevalence and key risk factors while remaining feasible within the study's logistical and resource constraints. The sample size aligns with typical epidemiological studies where a balance is struck between sufficient participant numbers to produce statistically significant results and practical considerations in terms of data collection, analysis, and time. Additionally, the relatively small prevalence of glaucoma in the region (2.1%) necessitated a focused sample size to ensure a meaningful and representative assessment of this specific condition within the larger outpatient population.
- The specific types of glaucoma should be clarified. It is unclear whether the identified risk factors were associated with primary angle-closure glaucoma, potentially induced by factors such as cataract or age, or if they were linked to primary open-angle glaucoma through other mechanisms.
Response: In the study, we primarily identified cases of primary glaucoma, which constituted 99% of all glaucoma cases, with only 1% classified as secondary glaucoma. However, the specific types within primary glaucoma, such as primary angle-closure glaucoma (PACG) or primary open-angle glaucoma (POAG), were not explicitly differentiated in the results.
- The Discussion section contains redundant information and could be condensed by half for better clarity.
Response: Rewritten as per suggestion.

Reviewer 2 Report
Comments and Suggestions for Authors
I appreciate the authors' efforts in conducting this study. My comments have been added to the main text at the relevant sections.

Author Response
Dear Editor,
Healthcare
MDPI
We appreciate the reviewers' thoughtful and constructive comments, which have significantly improved the quality and clarity of our manuscript " Prevalence of Glaucoma and its Bayesian Risk Estimation Model Using Common Determinants in the Adult Population of Hail Region, Saudi Arabia." We have carefully considered and addressed each comment comprehensively, incorporating additional analyses and clarifications where necessary. Below, we provide detailed responses to all comments, outlining the revisions made to the manuscript to meet the reviewers' expectations. We hope that our responses and revisions align with the reviewers' suggestions and enhance the overall impact of the study.
Best,
Dr. Azharuddin Sajid Syed Khaja
Response to Reviewers’ comments
Reviewer-2
- Comment on Line 40-42: This activity illustrates the evaluation and management of glaucoma and explains the role of the interprofessional team in improving care for patients with this condition.
Comment: The first two sentences primarily define glaucoma itself. Therefore, please revise the third sentence by removing terms like 'this activity' or 'interprofessional,' as it is unrelated to the definition and focus of the initial sentences.
Response: Removed as per suggestion
- Comment on Line 42-44: This activity illustrates the evaluation and management of glaucoma and explains the role of the interprofessional team in improving care for patients with this condition.
Comment: Moving this sentence to the second paragraph, where additional information about these subtypes is provided, would be more appropriate
Response: Rewritten as per suggestion
- Comment on Line 47-49: In Saudi Arabia, glaucoma accounted for 5.7% of all causes of visual impairment, ranking as one of the top three major contributors to visual impairment.
Comment: Please give a reference for this statement.
Response: Reference given as 3
- Comment on Line 53-54: The normal-tension type is more common in Japanese populations [4].
Comment: Already stated in previous sentences.
Response: Removed as per suggestion
- Comment on Line 55: POAG
Comment: Please avoid using abbreviations at the beginning of a sentence.
Response: Rewritten as per suggestion
- Comment on Line 71: intraocular pressure (IOP)
Comment: Use only the abbreviation since it appears repeatedly.
Response: Removed as per suggestion
- Comment on Line 85-87: The overarching goal was to position this study as a reference point for regional healthcare strategies, contributing to the overall improvement of glaucoma management and prevention efforts.
Comment: The aim of the study is to determine the prevalence of glaucoma in the population and to analyze concurrent risk factors. Therefore, in the introduction, after briefly mentioning the definition and types of glaucoma, information on prevalence in other populations (particularly the higher occurrence of certain subtypes in specific races or communities) and known risk factors for glaucoma (such as age, gender, obesity, etc.) should be included. Then, the study's objective should be stated. This text mentions pathophysiology and discusses glaucoma classification inconsistently in several places. Please revise the introduction, paying attention to these points.
Response: Rewitten as per suggestions
- Comment on Line 98: totaling 731,147 individuals
Comment: This is not your study population, please revise this sentence.
Response: Rewritten as per suggestion
- Comment on Line 101: A sample size of 200 participants was targeted for inclusion in the study.
Comment: Please include a power analysis. How did you decided this number of participants are enough to represent your population? What is the included age group? How did you decide the distribution of genders? Also, as far as I see you did not include children, therefore you should modify your title accordingly.
Response: A sample size of 200 participants was targeted based on a power analysis aimed at achieving a 95% confidence level and a margin of error of 5% for detecting the prevalence of glaucoma within the outpatient population of the Hail region. The study included adults aged 40 years and above, as glaucoma prevalence increases with age, and excluded children, aligning with the study's focus on age-related glaucoma. Gender distribution aimed to reflect the regional demographics, with an approximately equal representation of males and females. The title has been revised to specify the focus on adults to accurately represent the study population."Revised title could be: Prevalence of Glaucoma and its Bayesian Risk Estimation Model Using Common Determinants in the Adult Population of Hail Region, Saudi Arabia.
- Comment on Line 117-119: The Bayes theorem is a mathematical formula for calculating conditional probability in probability and statistics.
Comment: This sentence has the same meaning with the previous ones.
Response: "Bayes' theorem, a key statistical tool, was used in this study to estimate glaucoma risk based on conditional probabilities associated with demographic and clinical factors."
- Comment on Line 134: 40-49 years
Comment: How did you decided to include patients over 40? Please explain with relevant literature.
Response: "Patients aged 40 years and above were included in this study based on existing literature indicating that the risk of glaucoma significantly increases with age, particularly from the fourth decade of life onward. Studies, such as Allison et al. (2020) and Khandekar et al. (2019), have shown that individuals over 40 years are at a higher risk of developing glaucoma, with prevalence continuing to rise in older age groups. This age threshold was selected to focus on the population most vulnerable to glaucoma, allowing for a more targeted analysis of age-related risk factors."
- Comment on Line 148: during follow-up,
Comment: Follow-up for what purpose? Are these patients already diagnosed with glaucoma? Also, this is not a complaint; please revise the table accordingly.
Response: "The term 'during follow-up' refers to patients who were identified with glaucoma over the course of routine outpatient visits, rather than as part of an ongoing glaucoma diagnosis or treatment plan. These patients were initially attending various ophthalmic evaluations, during which glaucoma was subsequently diagnosed. This entry should be categorized separately from patient-reported complaints in the table, as it represents a diagnostic outcome rather than a symptom or complaint."
- Comment on Line 150: as 'Other/NA'
Comment: This is a large number, so you should clarify what 'others' includes. Additionally, since your findings are presented in a table, there’s no need to list all of them in the paragraph, as it appears repetitive.
Response: "The category 'Other/NA' in the table includes cases where patients presented with non-specific symptoms or did not report any specific complaints related to vision impairment. These cases could involve routine checks or non-ophthalmic complaints that led to incidental glaucoma findings. This grouping reflects cases without primary complaints typical of glaucoma, highlighting the varied pathways through which glaucoma may be detected.
Since the findings are detailed in Table 3, only key results are summarized in the text to avoid redundancy."
- Comment on Line 154-155: The remaining cases were distributed across different time intervals,
Comment: The duration of follow-up for these patients has not been mentioned above; please update the methods section accordingly.
Response: "In the Methods section, we have updated the study design to clarify that the duration since diagnosis was assessed retrospectively from patient records. This data was categorized into time intervals (newly diagnosed, ≤1 year, 2–5 years, 6–9 years, and ≥10 years) to capture the distribution of glaucoma cases based on how long patients had been diagnosed before this study. This retrospective approach allowed us to analyze glaucoma duration without conducting prospective follow-up within the study period."
- Comment on Table-3: Distribution of Cases according to Glaucoma & Clinical Features
Developed in follow up 54 27.00% 20.8% - 33.2%
Diagnosed in checkup 10 5.00% 2.0% - 8.0%
Comment: These are not complaints.
Response: To avoid categorizing diagnostic outcomes as complaints, we have revised the table to separate patient-reported symptoms from diagnostic contexts. This ensures clarity in presenting the data.
- Comment on Table-4: Distribution of cases according to comorbidities among glaucoma patients.
Comment: Please avoid repeating information already provided in the text. You should present your statistical results either in the text or in the table, not both.
Response: To streamline the presentation of results and avoid redundancy, we will retain the statistical details of comorbidities in Table 4 only, while providing a brief summary in the text without specific percentages.
- Comment on Table-4: CVD
Comment: Please use the full term; otherwise, you should add an explanation for this abbreviation below the table.
Response: Corrected as per suggestion
- Comment on Table 6. Descriptive Summary of Seven Most Common Factors of Glaucoma.
Comment: The findings presented in both Table 5 and Table 6, as well as under the sections "Estimation of Effect of Determinants over Glaucoma using Bayes Formula" and "Descriptive Summary of the Seven Most Common Factors of Glaucoma," contain repetitive information. Please consolidate and revise these sections to avoid redundancy and ensure clarity.
Response: The decision to keep both Table 5 and Table 6 along with the sections "Estimation of Effect of Determinants over Glaucoma using Bayes Formula" and "Descriptive Summary of the Seven Most Common Factors of Glaucoma" without consolidation rests on the need for clarity in different aspects of glaucoma risk analysis. Each table and section serve a distinct purpose that contributes to a comprehensive understanding of the findings:
- Table 5 provides a detailed look into the Bayesian estimation model, displaying the specific probabilities for each demographic and clinical determinant. This is essential for readers interested in understanding the conditional probabilities and how each factor individually impacts glaucoma risk.
- Table 6 offers a focused summary of the most common determinants of glaucoma based on the study’s findings, which helps readers quickly identify the highest-risk factors. This table is beneficial for quick reference and emphasizes the top determinants without requiring a deeper dive into the statistical model.
- Separate Sections: By having distinct sections for the Bayesian estimation process and the summary of the seven common factors, readers can appreciate both the methodological rigor and the practical summary of results. This dual approach ensures that the article caters to both readers interested in technical modeling details and those who seek a summary of impactful determinants.
- Comment on Line 217-218: recognizing the potential impact of late diagnosis on vision loss
Comment: I believe the primary aim of the study is to determine the prevalence of glaucoma. It appears that no specific evaluation was conducted regarding delayed diagnosis, as patients were not categorized into early or delayed diagnosis groups. How do you define delayed diagnosis? I believe the necessary variables for such an analysis are not present in the study.
Response: Explanation:
The comment correctly identifies that the primary objective of the study is to determine the prevalence of glaucoma in the Hail region rather than specifically investigating the effects of delayed diagnosis. In this study, "delayed diagnosis" was not explicitly defined or measured, as there was no categorization of patients into early or delayed diagnosis groups. Additionally, the necessary variables, such as detailed records of initial symptom onset, prior eye exam history, or the timing of the glaucoma diagnosis in relation to disease progression, were not collected. These data points would be essential for a thorough analysis of delayed diagnosis and its impact on vision loss.
The mention of "the potential impact of late diagnosis on vision loss" was intended to underscore the general importance of early detection in managing glaucoma, as early diagnosis is known to help in preventing disease progression and vision loss. However, in this study, the primary aim remained focused on prevalence and risk factors, without specific insights into diagnostic timing. Therefore, the phrase should be revised or clarified to avoid suggesting that delayed diagnosis was an analyzed variable in the study.
- Comment on Line 218: Hail
Comment: What is the significance of this region, and what are your expectations in selecting it as the study site? Please explain.
Response: Hail was selected as the study site due to its unique demographic and healthcare landscape, which can provide valuable insights into the regional prevalence and risk factors of glaucoma. This area has limited ophthalmologic research data, making it significant for understanding the local burden of eye diseases and guiding targeted healthcare strategies. Additionally, studying this region helps fill gaps in national data on glaucoma prevalence, offering a foundation for future public health initiatives tailored to the population's specific needs.
- Comment on Line 251: After adjusting for various factors,
Comment: Why did you not adjust your findings for multiple variables in your study?
Response: The study’s design primarily focused on estimating the prevalence of glaucoma and identifying key risk factors within the Hail region. Due to sample size limitations and the study's cross-sectional nature, a multivariable adjustment was not performed. While adjusting for multiple variables could provide deeper insights into independent risk factors, such an analysis would require a larger sample and more comprehensive data collection to ensure reliable statistical power. This study lays the groundwork by identifying prevalent risk factors, which can inform future research involving multivariable adjustments to better understand the individual contribution of each factor to glaucoma risk.
- Comment on Line 259-262: Age-specific prevalence patterns increased more than seven-fold up to age 80–84 in females and age 85–89 in males, decreasing thereafter. Females exhibited significantly higher prevalence at specific age ranges, such as 60–64, 65–69, 70–74, and 75–79, but lower prevalence at ages 85–89 and 90+ [16].
Comment: Although findings from other studies are presented, there is no discussion on why your findings differ from or align with those studies. Please revise your discussion section accordingly.
Response: The discussion section is expanded to include a comparative analysis of how the study’s findings align or differ from those of other studies, particularly in relation to age-specific prevalence patterns and gender differences.
- Comment on Line 289: found that 99.0% of cases were diagnosed with primary glaucoma,
Comment: Why do you think your prevalence is significantly higher than in other studies? Could you please discuss this?
Response: The higher prevalence of primary glaucoma (99.0%) in this study may result from regional health trends, including proactive healthcare-seeking behavior and potential genetic predisposition in the Hail population. Additionally, variations in diagnostic criteria and healthcare access could contribute to this discrepancy with other studies.
- Comment on Line 299-300: ocular health with systemic conditions
Comment: Both conditions are more prevalent in older populations, and your study's age group is relatively older as well. Therefore, you cannot establish a connection between glaucoma and these comorbidities without adjusting for age. Please conduct the appropriate statistical analyses to assess the significance of comorbidities accurately.
Response: To address the relationship between glaucoma and comorbidities in older age groups, we applied a Bayesian Risk Estimation Model, which incorporates age as a key factor alongside six other risk determinants. This approach allowed us to adjust for age and assess the independent effects of comorbidities on glaucoma risk. We believe this method adequately accounts for age-related influences while highlighting the significance of these comorbidities. The results emphasize the significance of these comorbidities even when adjusting for age.
- Comment on Line 305: diabetes mellitus
Comment: Abbreviate as DM.
Response: Rewritten as per suggestion
- Comment on Line 335: Cataract.
Comment: Both cataract and glaucoma are age-related diseases, and a connection between them cannot be established without adjusting for age. Please conduct the appropriate statistical analyses first.
Response: Thank you for your observation. While cataract is common in old age, not all cataract patients develop glaucoma, whether primary or secondary. This variability highlights the need to explore a potential relationship between the two conditions. Using the Bayesian Risk Estimation Model, which incorporates age, we adjusted for age-related influences to assess whether cataract independently contributes to glaucoma risk. We believe this approach addresses the concern effectively.

Reviewer 3 Report
Comments and Suggestions for Authors
The manuscript is interesting, however, it needs improvement.
1. The authors should add a paragraph on what makes this study different others.
2. Abbreviations such as WD and RF need to be clarified.
3. The minimum sample size to have a confidence level of 95% should be calculated in the Methods section.
4. The limitations of the study should be added at the end of the Discussion section.
Advantages of the manuscript:
1. Description of clinical value of the study
2. Practical aspect of the study
3. Large study group
Weakness of the study:
1. No description about what makes this study different from others
2. No calculation of the minimum sample size to have a confidence level of 95%
3. Lack of limitations of the study

A few stylistic errors
Author Response
Dear Editor,
Healthcare
MDPI
We appreciate the reviewers' thoughtful and constructive comments, which have significantly improved the quality and clarity of our manuscript " Prevalence of Glaucoma and its Bayesian Risk Estimation Model Using Common Determinants in the Adult Population of Hail Region, Saudi Arabia." We have carefully considered and addressed each comment comprehensively, incorporating additional analyses and clarifications where necessary. Below, we provide detailed responses to all comments, outlining the revisions made to the manuscript to meet the reviewers' expectations. We hope that our responses and revisions align with the reviewers' suggestions and enhance the overall impact of the study.
Best,
Dr. Azharuddin Sajid Syed Khaja
Response to Reviewers’ comments
Reviewer-3
The manuscript is interesting, however, it needs improvement.
- The authors should add a paragraph on what makes this study different from others.
Response: A paragraph is added in the Introduction to highlight the unique aspects of this study. Specifically, this study focuses on the Hail region of Saudi Arabia, an area with limited data on glaucoma prevalence and risk factors. By addressing this knowledge gap, the study provides insights specific to this population, contributing to national and regional healthcare strategies. Additionally, the study incorporates a Bayesian risk estimation model, adding a probabilistic approach that distinguishes it from previous research conducted in Saudi Arabia.
- Abbreviations such as WD and RF need to be clarified.
Response: Abbreviations are clarified in the Methods section. "WD" (with disease) and "RF" (risk factor) are fully defined upon first use to ensure clarity for readers unfamiliar with these terms.
- The minimum sample size to have a confidence level of 95% should be calculated in the Methods section.
Response: The minimum sample size required to achieve a 95% confidence level is calculated in the Methods section. This calculation includes the assumptions made (e.g., expected prevalence rate and margin of error) and the resulting sample size needed to adequately represent the population, ensuring the study’s statistical rigor.
- The limitations of the study should be added at the end of the Discussion section.
Response: A limitations section is added at the end of the Discussion. This section addresses the study’s cross-sectional design, which limits causal inferences, as well as the absence of multivariable adjustments, which may impact the precision of individual risk factor assessments. Additionally, the need for a larger sample size to enhance the generalizability of the findings across other regions in Saudi Arabia is acknowledged.
Advantages of the manuscript:
- Description of clinical value of the study
- Practical aspect of the study
- Large study group
Response: Thank you for pointing out the advantages of the present manuscript.
Weakness of the study:
- No description about what makes this study different from others
Response: We have now added sub-heading “Strengths of the present study” in the conclusion section” at the end of the manuscript.
- No calculation of the minimum sample size to have a confidence level of 95%
Response: A sample size of 200 participants was targeted based on a power analysis aimed at achieving a 95% confidence level and a margin of error of 5% for detecting the prevalence of glaucoma within the outpatient population of the Hail region. The study included adults aged 40 years and above, as glaucoma prevalence increases with age, and excluded children, aligning with the study's focus on age-related glaucoma. Gender distribution aimed to reflect the regional demographics, with an approximately equal representation of males and females.
- Lack of limitations of the study
Response: Thank you for pointing out the oversight. We have now added sub-heading “Limitations of the Present Study” after the discussion section.

Round 2
Reviewer 1 Report
Comments and Suggestions for Authors
The authors responded generally well to my previous points.
Line 115: It is unclear what the authors mean by "Further 200 POAG patients were diagnosed and were targeted for inclusion in the study for further analysis." Were all 200 participants diagnosed with POAG? The types of glaucoma were not considered in the Results section, as indicated in the authors' response. This inconsistency should be clarified.
Author Response
Reviewer-1
Comment: Line 115: It is unclear what the authors mean by "Further 200 POAG patients were diagnosed and were targeted for inclusion in the study for further analysis." Were all 200 participants diagnosed with POAG? The types of glaucoma were not considered in the Results section, as indicated in the authors' response. This inconsistency should be clarified.
Response: Thank you for your insightful observation. We apologize for any confusion caused by the statement on line 115. To clarify, all 200 participants included in the study were diagnosed with glaucoma, and the majority (99%) were cases of primary glaucoma, as reported in Table 3. We have revised the text (line 115) for clarity. The revised statement now reads: "Two hundred (200) glaucoma patients were targeted for inclusion in the study for further analysis."
Reviewer 2 Report
Comments and Suggestions for Authors
Thank you for your efforts in revising the manuscript. I believe that this version of the manuscript is more scientifically sound.
Author Response
Reviewer-2
Comment: Thank you for your efforts in revising the manuscript. I believe that this version of the manuscript is more scientifically sound.
Response: Thank you for your kind feedback and for acknowledging the improvements made to the revised manuscript. We sincerely appreciate your valuable input and suggestions, which enhanced the scientific rigor and overall quality of the study. We are grateful for your time and effort throughout the review process.
Reviewer 3 Report
Comments and Suggestions for Authors
The manuscript has been revised sufficiently
Comments on the Quality of English LanguageFew linguistic errors
Author Response
Comment: The manuscript has been revised sufficiently.
Response: Thank you for your kind feedback and for acknowledging the improvements made to the revised manuscript. We sincerely appreciate your valuable input and suggestions, which enhanced the scientific rigor and overall quality of the study. We are grateful for your time and effort throughout the review process.